# Early forecasting of tsunami inundation from tsunami and geodetic observation data with convolutional neural networks

Fumiyasu Makinoshima [1✉], Yusuke Oishi [1], Takashi Yamazaki[1], Takashi Furumura [2] & Fumihiko Imamura [3]

Rapid and accurate hazard forecasting is important for prompt evacuations and reducing casualties during natural disasters. In the decade since the 2011 Tohoku tsunami, various tsunami forecasting methods using real-time data have been proposed. However, rapid and accurate tsunami inundation forecasting in coastal areas remains challenging. Here, we propose a tsunami forecasting approach using convolutional neural networks (CNNs) for early warning. Numerical tsunami forecasting experiments for Tohoku demonstrated excellent performance with average maximum tsunami amplitude and tsunami arrival time forecasting errors of ~0.4 m and ~48 s, respectively, for 1,000 unknown synthetic tsunami scenarios. Our forecasting approach required only 0.004 s on average using a single CPU node. Moreover, the CNN trained on only synthetic tsunami scenarios provided reasonable inundation forecasts using actual observation data from the 2011 event, even with noisy inputs. These results verify the feasibility of AI-enabled tsunami forecasting for providing rapid and accurate early warnings.

[1] Fujitsu Laboratories Ltd., Kawasaki, Japan. [2] Earthquake Research Institute, The University of Tokyo, Bunkyo-ku, Tokyo, Japan. [3] International Research Institute of Disaster Science (IRIDeS), Tohoku University, Sendai, Japan. ✉email: f.makinoshima@fujitsu.com

The rapid forecasting of hazards and dissemination of warnings can increase evacuation lead times and thus are key to saving lives during natural disasters. For tsunami disasters, quick evacuations supported by such warnings can drastically reduce the number of casualties, but inaccurate hazard forecasts and warnings can have the opposite effect. During the 2011 Tohoku tsunami event, the earthquake magnitude and tsunami height were initially underestimated based on a pre-computed database; as a result, some residents felt safe based on the initial warning and were unaware of the need for evacuation[1]. Although the warning was updated several times based on additional observations, the updated information could not always reach the residents due to communication disruptions; consequently, many of the coastal residents did not realise the tsunami risk at their locations. The underestimation of tsunami risk increased the number of tsunami-induced casualties, and as a result, Japan experienced the loss of over 18,000 citizens, even with the in-place warning system. The need for accurate and reliable early warnings has been common during past mega-tsunamis; notably, the 2004 Indian Ocean tsunami had a catastrophic regional impact[2], and the immense loss of nearly 230,000 lives stressed the importance of tsunami early warning systems, resulting in efforts to establish tsunami early warning frameworks in broader regions[3]. Therefore, fast and accurate tsunami forecasting methods based on real-time tsunami observation data are urgently needed, and the early warnings provided can contribute to mitigating casualties in future tsunami events.

To date, tsunami early warning systems have been developed based on past tsunami catastrophes and available technologies[4]. Especially in the decade since the 2011 Tohoku tsunami, dense tsunami observation networks have been implemented[5,6], and various tsunami forecasting methods using real-time observation data, such as real-time tsunami inundation simulations using supercomputers[7] with rapid source estimations[8,9] and data assimilation approaches[10,11], have been proposed based on the lessons learned from the 2011 event. However, real-time tsunami inundation forecasting immediately after an earthquake has remained challenging due to the difficulty of rapid estimation of the tsunami source, in which various uncertainties exist[12], and due to the high computational costs associated with simulating nonlinear tsunami propagation in shallow water.

To overcome the above challenges, we present a tsunami forecasting method using a convolutional neural network (CNN) developed as a deep learning approach in AI research. The present method is capable of directly forecasting tsunami inundation based solely on up-to-date observation data and does not require extensive computational resources, such as those provided by supercomputers. During the past decade, deep learning has achieved great success in image and pattern recognition[13] as well as in broader areas, including physics-based simulations such as structural analysis[14] and computational fluid dynamics[15]; moreover, tsunami observation networks have been improved.

In this work, we utilise a CNN to process valuable data from dense tsunami and geodetic observation networks and achieve remarkable tsunami forecasting performance. A notable advantage of a CNN is its low computational cost; i.e., the computational cost of CNN inference is much lower than that of nonlinear tsunami propagation simulations. Additionally, the present approach does not require a tsunami source estimation process since our CNN is designed for end-to-end forecasting from observation data to tsunami inundation forecasting. Therefore, the present CNN can immediately and accurately predict the tsunami inundation time series at a single location. To the best of our knowledge, this study is the first attempt at end-to-end tsunami inundation forecasting with a CNN, and the results verify the feasibility of AI-enabled tsunami forecasting for the establishment of early warnings.

## Results

**CNN tsunami forecasting**. Figure 1 shows a schematic view of the proposed tsunami forecasting method based on a CNN. First, 10,000 cases of numerical tsunami propagation and inundation simulations solving a set of partial differential equations were conducted to prepare the data sets for training the CNN. In the simulations, sets of synthetic observation data at observation points and the resulting tsunami inundation waveforms within 2 h were calculated based on randomly generated tsunami scenarios. For the tsunami observations obtained with ocean bottom pressure gauges, simulated tsunami waveforms converted into pressure waveforms at the ocean bottom were used as inputs. Then, we trained a CNN with synthetic tsunami data to directly predict the tsunami inundation waveform at a single onshore location (the green star in Fig. 1) solely from the observations. We built a 1-D CNN comprising a total of 15 layers for tsunami forecasting since compact 1-D CNNs are suited for real-time and low-cost applications because of their low computational complexity[16]. The stacked waveforms were fed into the network as inputs, and the corresponding features were extracted via convolutional and pooling-like layers. To consider additional observations as inputs for the network, we simply stacked same-size arrays so that the size of the input channels equalled the considered number of observation points. Since recent studies on tsunami source inversion suggest that inland geodetic observational data are useful for determining the tsunami source[8,9,17], we also considered inland geodetic observations of initial ground heights as additional inputs to the CNN. The geodetic observations were fed as vectors with the same length as the offshore waveforms. The onshore tsunami inundation waveform was then forecasted by the fully connected layers using the features extracted from the convolutional layers. In contrast to general time series forecasting using deep learning, in which the subsequent transition of values at future time $t + h$ is predicted based on the available observations at time $t$ in the same series[18], our network predicts a future onshore tsunami inundation waveform based on the available observations at different offshore and onshore points. After the training process, the trained CNN can output a tsunami inundation waveform at a single location on land (the green star in Fig. 1) when observation data for an unknown tsunami are given. Preparing multiple networks enables tsunami forecasting for multiple points.

**Data preparation for the CNN**. The source fault geometry of the 2011 Tohoku-Oki earthquake was considered based on the parameters presented in Fujii et al.[19], and 44 sub-faults were used to generate synthetic tsunami data sets for the training of CNN (Fig. 2a). Here, the slip parameters of each sub-fault were randomly assigned within their defined range for the sub-faults to generate various tsunami scenarios. We conducted tsunami simulations for a total of 12,000 scenarios and obtained observations for inputs (red points in Fig. 2a) and an onshore tsunami inundation waveform at a single location (the green star in Fig. 2b) to train the CNN. A total of 49 offshore tsunami observation points and five onshore geodetic observation points were considered as inputs for the CNN based on an actual tsunami observation network[20] and the Global Navigation Satellite System observation network[21] in Japan. These tsunami and geodetic observation points were selected to cover the fault region and the forecasting point. The offshore tsunami observation waveforms were sampled at 1 Hz, and a constant deformed ground height with the same number of samples was considered

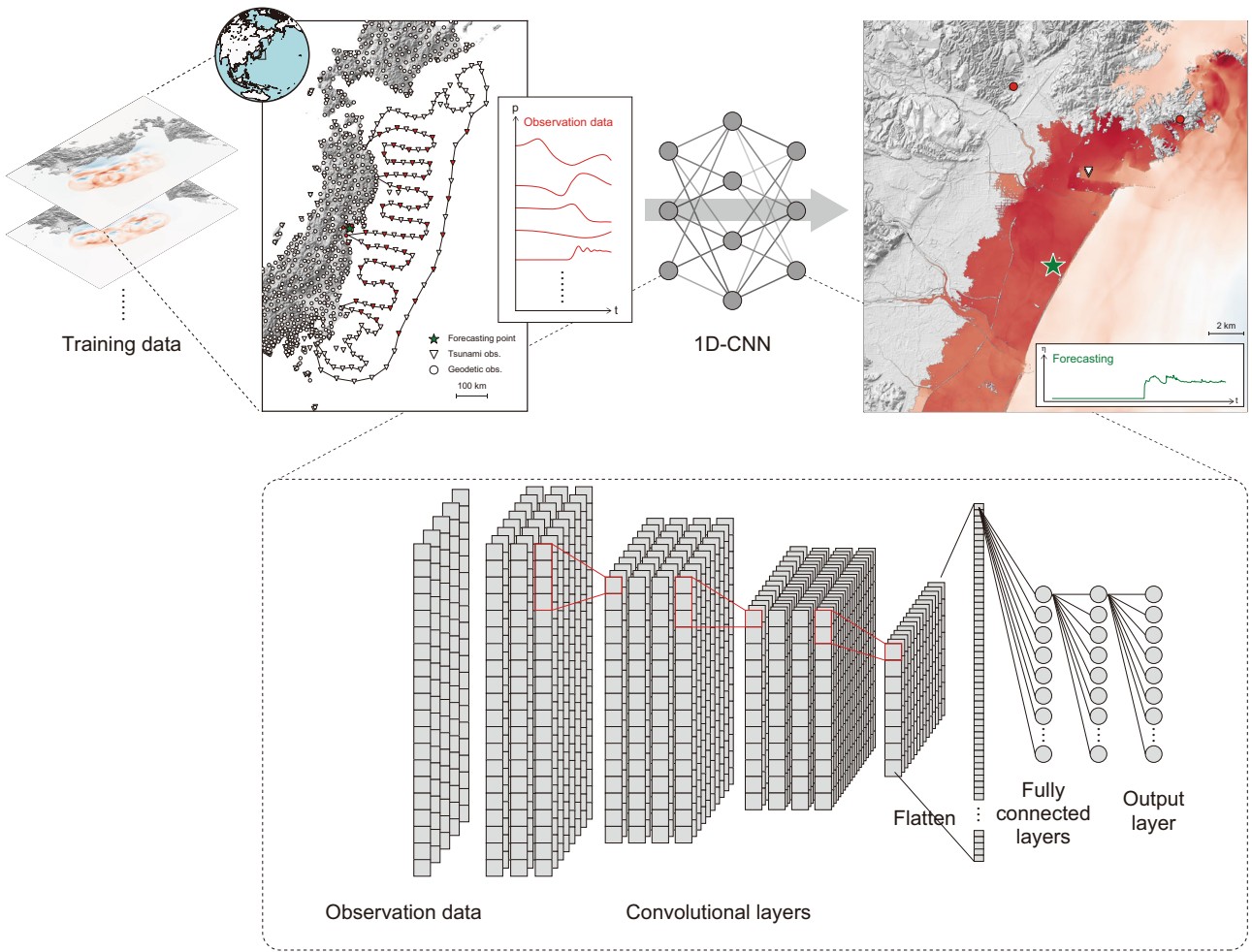

**Fig. 1 Schematic view of tsunami forecasting by the convolutional neural network (CNN).** The CNN learns the relation between observation data and the resulting tsunami waveform from thousands of numerical simulation results. The tsunami and geodetic observation points are uniformly selected to cover the forecasting areas and are illustrated as red points in the map. After the training process is completed, the trained CNN can predict a time series of inundation tsunami waveform at the onshore forecasting site, which is represented with a green star, solely from observation data for unknown tsunami scenarios.

as the onshore geodetic observations. The forecasting waveform was sampled at 0.5 Hz with a data size of 3600. For the observation data, various time windows of tsunami observations (e.g., 5, 10, 15, 20, 25, and 30 min) were considered as inputs for the CNN to investigate the effect of the observation length on the forecasting accuracy. From the 12,000 simulation results, we used 10,000 cases for training, 1000 cases for validation monitoring, and 1000 for testing. The generated earthquake scenarios have a seismic moment $M_0$ ranging between $4.03 \times 10^{22}$ and $8.21 \times 10^{22}$ Nm, which corresponds to a moment magnitude $M_w$ ranging from 9.0 to 9.2, assuming a rigidity of 30 GPa (Fig. 2c). The initial sea-bottom deformation caused by an earthquake was calculated with Okada's formula[22], and the tsunami propagation and inundation were simulated using TUNAMI-N2 code (see "Methods" for details). A single tsunami simulation for 2 h to generate data for the CNN required ~3 h using a CPU node with two Intel Xeon Gold 6148 processors with 384 GiB memory.

**Network configuration and training.** The constructed CNN consisted of nine convolutional layers and three pooling-like convolutional layers followed by three fully connected layers (Supplementary Table 1). Each convolutional layer was followed by a Leaky ReLU activation function[23] with a negative slope

parameter of $a = 0.01$. Dropout with a dropout ratio of $p = 0.5$ was applied to the outputs of the fully connected layers to prevent overfitting[24]. In the convolutional layers, we set the kernel size, stride and padding to minimise changes in the array size, and dimensionality reduction was performed mainly by the pooling-like layers. For the 5, 15, 25 and 35 min observation cases, the kernel size of the last convolutional layer was set as 3 to prevent the remainder from being generated, but such changes were minimised to evaluate the effect of the observation lead time. Thus, a longer observation period leads to a greater number of learnable parameters in the network.

We considered the mean squared error (MSE) between the simulated and forecasting waveforms over the forecasting time as the loss function. The network parameters were optimised to minimise the loss function by the Adam optimisation algorithm[25]. We used the default parameters suggested in the original paper that proposed the Adam algorithm ($\beta_1 = 0.9$, $\beta_2 = 0.999$ and $\varepsilon = 10^{-8}$), except for a step size of $\alpha = 10^{-4}$. The training of the network was performed in the AI Bridging Cloud Infrastructure, a GPU-accelerated supercomputer in which each node has two Intel Xeon Gold 6148 CPUs and four NVIDIA Tesla V100 SXM2 GPUs with 384 GiB memory. The network training and validation in this study were implemented using Pytorch[26] and Horovod[27]. The batch size for each GPU during

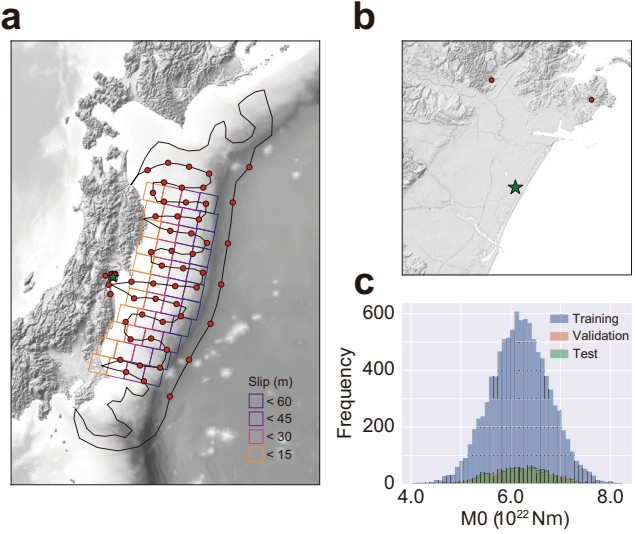

**Fig. 2 Fault and observation settings for generating data sets for CNN. a** Fault geometry and observation point as input for convolutional neural network (CNN). Red circles represent observation points for CNN. A total of 49 offshore tsunami observation points and five onshore geodetic observation points were considered to cover the fault region and the forecasting point (the green star). Fault geometry is considered based on Fujii et al.[19], but its slip amount is randomly assigned within a given range to generate various tsunami scenarios. **b** A close view of the forecasting site, the Sendai plain where experienced the large extent of tsunami inundation. The green star is the forecasting site for which the CNN forecast a tsunami inundation waveform. **c** Distribution of the seismic moment of generated earthquake scenarios. A total of 12,000 scenarios were generated and divided into 10,000 training sets, 1000 validation sets and 1000 test sets.

the training phase was 25, and five computer nodes were used for training. We considered 3000 epochs in the training process and retained the model that yielded the minimum validation loss for the validation data sets during training. The training process was completed within 2 h even for the largest network in this study.

**Forecasting performance on synthetic tsunamis**. We evaluated the performance of the trained CNN by analysing the forecasting results for 1000 test tsunami scenarios that were not considered during the training process, and we confirmed that the CNN successfully predicted the tsunami inundation waveform at the forecasting site (Fig. 3b). For the evaluation of the forecasting accuracy, we considered two metrics: the maximum tsunami amplitude, defined as the maximum value of the tsunami amplitude over the forecasting time, and the tsunami arrival time, defined as the time when the tsunami flow depth first exceeds 10% of the maximum flow depth. Even with only 5 min offshore tsunami and inland geodetic observations, the mean absolute errors of the maximum tsunami amplitude and the tsunami arrival time were 0.4 m and 47.7 s, respectively. The average relative errors were 8.1% and 1.2% for the maximum tsunami amplitude and the tsunami arrival time, respectively. For these forecasts, the trained CNN required only 0.004 s on average using a single CPU node with 40 cores; this approach is much faster than conventional simulation-based forecasting approaches and requires fewer computational resources. The trained CNN yielded accurate and rapid forecasts of both the tsunami size and the arrival time for various tsunami scenarios directly from the observations.

**Effects of the offshore observation length and geodetic data**. We investigated the effect of the length of the observation inputs

(5, 10, 15, 20, 25 and 30 min) and the importance of geodetic observation inputs by training different CNN models with different inputs (Fig. 4). The results show that longer observations led to higher forecasting accuracy. Using the geodetic observation data, the CNN achieved good forecasting performance equivalent to that of a CNN with a long observation period. We confirmed that this performance improvement achieved with comparatively long-term observations and geodetic data corresponds mainly to increased accuracy in the initial ground height estimation in which the CNN with a short observation length and without geodetic data exhibited poor accuracy (Supplementary Fig. 1). This result can be explained by the characteristics of tsunami long wavelength. Since the propagation speed of a tsunami is slow in shallow water, it is difficult to obtain the waveforms generated from nearshore faults that cause coastal subsidence with a short observation length; thus, short-term observations alone are not sufficient for estimating the magnitude of subsidence. Inverting the offshore fault slip distribution from onshore geodetic information can lead to non-unique solutions; however, geodetic data offer direct information about subsidence much faster than offshore observations. The proposed CNN integrated different information types from different observations and achieved the presented forecasting performance, even with very short observation periods.

**Sensitivity of offshore and onshore observations**. To understand information processing in CNN tsunami forecasting models, we conducted a sensitivity analysis (e.g., occlusion test[28,29]) of the trained CNN models. In this analysis, we systematically removed inputs from certain observation points and evaluated the resulting amounts of change in the forecasting results to represent the impact on the CNN. The sensitivity analysis was conducted for models with different observation lengths, and the sensitivity of the observation points for forecasting was visualised (Fig. 5). The observation points with high sensitivities were located in the specific region of the observation network along the major path of tsunamis towards the forecasting site. High sensitivities were observed mainly over large slip areas since the amount of slip on an offshore fault has a predominant influence on tsunami inundation. In contrast, the information from distant observation points had almost no effect on the forecast, and thus, this information was not important for the CNN. This result indicates that for an accurate forecast, the CNN requires only certain observation points along the path of a tsunami propagating to a forecasting site. As the observation time increases, high sensitivities can also be confirmed at nearshore tsunami observation points. The CNNs using both offshore tsunami and onshore geodetic observations showed high sensitivities at both onshore and offshore points. The increased sensitivities for nearshore tsunami observations with longer observation periods and the higher sensitivities for additional onshore geodetic observations suggest that the CNN effectively integrates available information within limited observation periods to achieve high forecasting accuracy.

**Forecasting speed**. The computational time for tsunami forecasting with the CNN was measured to investigate the forecasting speed and assess the ability of the method to be employed for the issuance of tsunami warnings. Here, we measured the time required to forecast 1000 test scenarios using a single CPU node with 40 cores. Table 1 reports the average computational time for tsunami forecasting. The computational time increases as the observation time increases since a consistent CNN architecture is adopted for all observation lengths; a large neural network structure and a corresponding increase in the number of

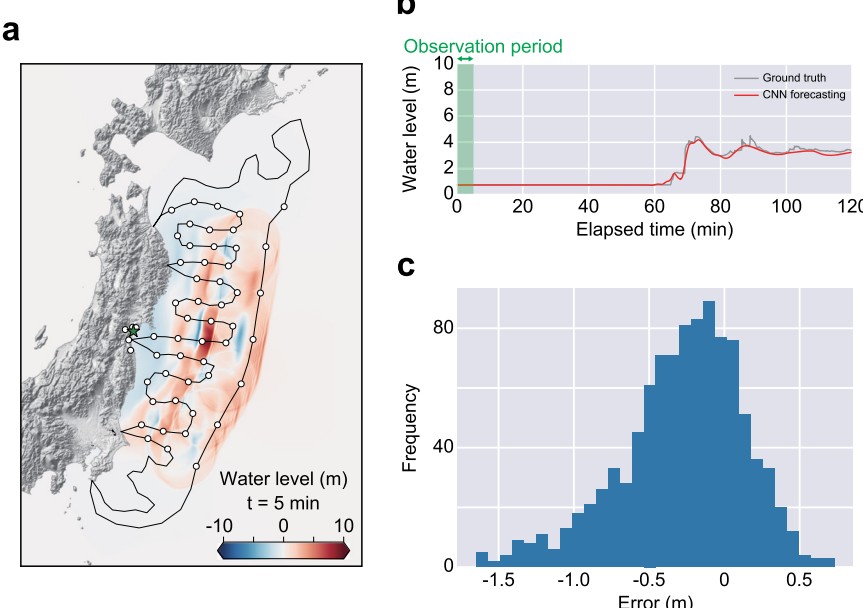

**Fig. 3 Application of the trained convolutional neural network (CNN) to 1000 test cases (5-min offshore and geodetic observations). a** Observation points and a snapshot of tsunami propagation at 300 s for a simulated test scenario. The location of the observation points as inputs for the CNN are illustrated as circles. The forecasting site is represented with the green star. **b** Result of tsunami inundation forecasting with the CNN at the forecasting site with the ground-truth simulation result. **c** Error distribution of the maximum tsunami amplitude at the forecasting site for 1000 test scenarios.

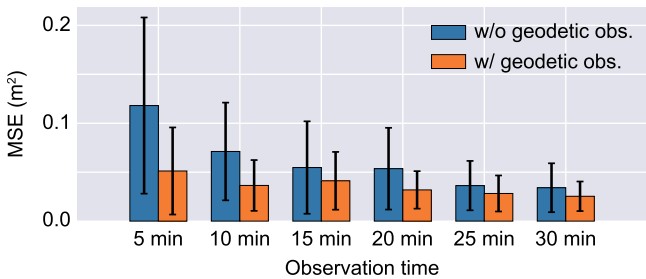

**Fig. 4 Mean squared error (MSE) in the convolutional neural network (CNN) tsunami forecasts for 1000 test scenarios with different observation lengths with or without geodetic data.** The height of the bar represents the mean value for 1000 test scenarios, and the error bar in the figure represents the standard deviation.

parameters are needed for longer observation times. The addition of 5 min observations increased the number of parameters by ~13 million, mainly due to the larger size of the fully connected layer after the convolutional layers. Nevertheless, the computational time for tsunami forecasting with the CNN was only 0.011 s, even for the largest CNN settings in the test (30 min offshore tsunami observations with geodetic observations). The addition of five geodetic input channels slightly increased the number of parameters by 1920 but had a negligible effect on the computational time. Thus, for CNN tsunami forecasting, the use of geodetic data was effective from the perspectives of not only the forecasting accuracy but also the computational time required to consider additional inputs. The tsunami forecasting speed achieved by the CNN is sufficiently fast to provide tsunami warnings, even with limited computational resources.

**Application to the 2011 Tohoku tsunami event.** We trained the CNNs using the 10,000 synthetic tsunami scenarios with the observation settings at the time the 2011 Tohoku tsunami event

occurred and investigated the forecasting performance of the CNN for real events using real-world data (Fig. 6a). This application used the same network configuration and the same 10,000 tsunami scenarios employed in the previous tests using synthetic data. We used publicly available observation data[30,31] during the event as inputs for the CNN (Fig. 6b, c), i.e., three offshore tsunami observations recorded by GPS buoys (803, 801 and 806) and three onshore GNSS observations (Rifu, Watari and Souma1). For the missing parts in the tsunami waveforms observed by GPS buoys, 1 Hz data were prepared by cubic interpolation. For the geodetic observations, displacements at 5 min after the occurrence of the earthquake were used as inputs. Since complete data were not available at Souma1, we used the latest observation as the input. The forecasting site is shown as the green star in Fig. 6a. Most of the buildings around the forecasting site were totally destroyed by the tsunami; however, Arahama Elementary School (Arahama ES, illustrated as the black cross in Fig. 6a) located close to the forecasting site provided survey results to verify the inundation forecast. During the 2011 event, the tsunami reached the second floor of the Arahama ES[32], and a survey immediately after the event reported that tsunami debris were found at a height of 4.62 m above the school basement[33]. The arrival time of the devastating tsunami at this site was also estimated as ~15:55 (JST) based on a stopped clock[34].

We trained the CNNs using only synthetic tsunami data with different observation periods and forecasted the tsunami inundation waveform using the actual observation data as inputs for the CNNs (Fig. 6d). Initially, the CNNs forecasted unphysical waveforms with 20 min or less observations, when almost no tsunami signals were available. After obtaining the first positive peak of the tsunami from 30 min observations, the CNN forecasted an inundation waveform, but the forecasted amplitude was small compared to the actual trace at Arahama ES. With the 35 min observations, in which the entire first positive peak of the tsunami is available, the CNN forecasted a maximum flow depth of 3.88 m at 3974 s after the earthquake. Further 40 min observation revealed the negative peak of the tsunami, enabling

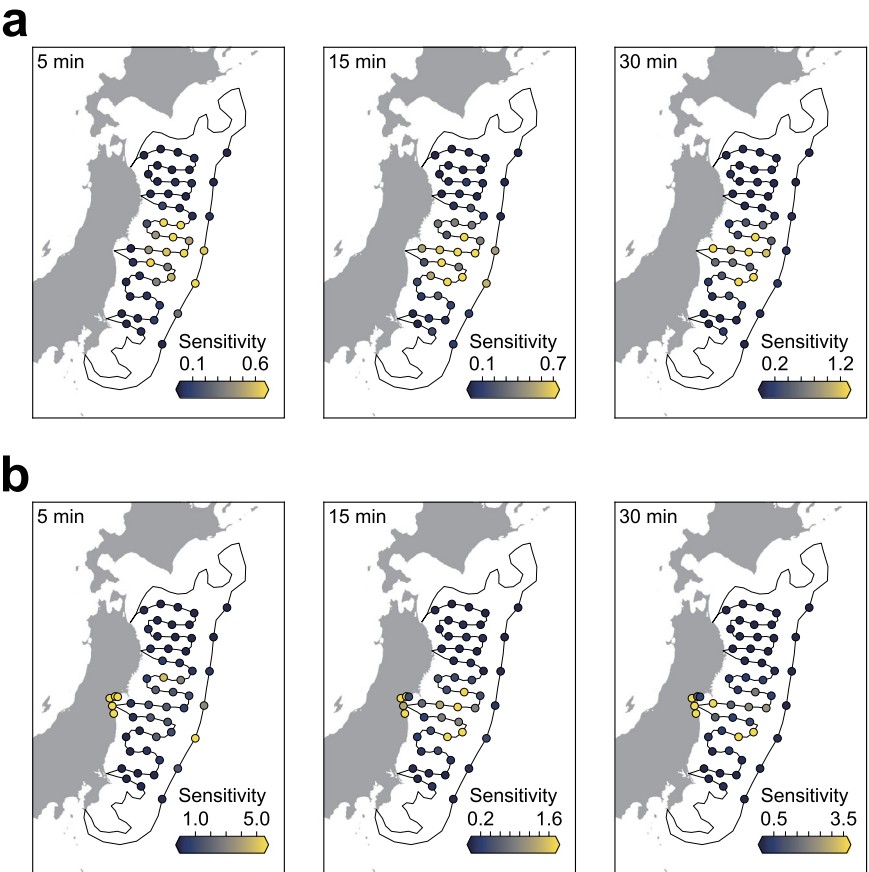

**Fig. 5 Results of sensitivity analysis for convolutional neural networks (CNN) with different observation times. a** CNN trained only with offshore tsunami observation data. **b** CNN trained with both offshore tsunami and onshore geodetic observation data. The observation points with high sensitivity indicate that the forecasting result changes considerably when the inputs from these observation points are lacking.

| Table 1 Computational time for tsunami forecasting. | | |
|---|---|---|
| **Observation time and data** | **Number of learnable parameters** | **Computational time/scenario (s)** |
| 5 min w/o geodetic data | 45,186,960 | 0.004568 |
| 10 min w/o geodetic data | 58,752,912 | 0.005555 |
| 15 min w/o geodetic data | 71,401,360 | 0.008212 |
| 20 min w/o geodetic data | 84,967,312 | 0.009629 |
| 25 min w/o geodetic data | 97,615,760 | 0.010030 |
| 30 min w/o geodetic data | 111,181,712 | 0.011222 |
| 5 min w/ geodetic data | 45,188,880 | 0.004487 |
| 10 min w/ geodetic data | 58,754,832 | 0.005436 |
| 15 min w/ geodetic data | 71,403,280 | 0.008063 |
| 20 min w/ geodetic data | 84,969,232 | 0.009101 |
| 25 min w/ geodetic data | 97,617,680 | 0.010226 |
| 30 min w/ geodetic data | 111,183,632 | 0.011086 |

the forecast of a larger inundation depth (5.64 m at 3952 s). Although the reported tsunami traces do not directly reflect the actual maximum flow depth, the CNN forecasting can be considered reasonably accurate. However, the forecasted arrival time of the maximum flow depth was ~3 min earlier than the estimated arrival time indicated by the stopped clock at the Arahama ES.

To further verify the CNN forecasting result, we trained the CNN for forecasting a tsunami waveform at the Sendai New Port where a time series of the tsunami until the first peak was recorded by a wave gauge (subsequent observations were not available because the gauge was destroyed by the tsunami). The forecasting results for the Sendai New Port are summarised in

Fig. 6e. Similar to the previous inundation forecasting results, the CNN could not provide a reasonable forecast with 20 min or less observation time when sufficient tsunami signals were not available. After obtaining the positive peak of the tsunami using 30 min observation, the CNN could forecast the tsunami having the first peak value of 6.78 m, which is compatible with the observed first peak (6.62 m). The CNNs trained with 35 and 40 min observations forecasted similar peaks (5.77 and 6.42 m, respectively), and the rise of the first wave was more consistent with the observation data. Nevertheless, even with sufficient offshore tsunami observations, an ~3 min difference between the observation and the forecast appeared in the tsunami arrival time, suggesting that the CNN forecast tends to be slightly earlier than

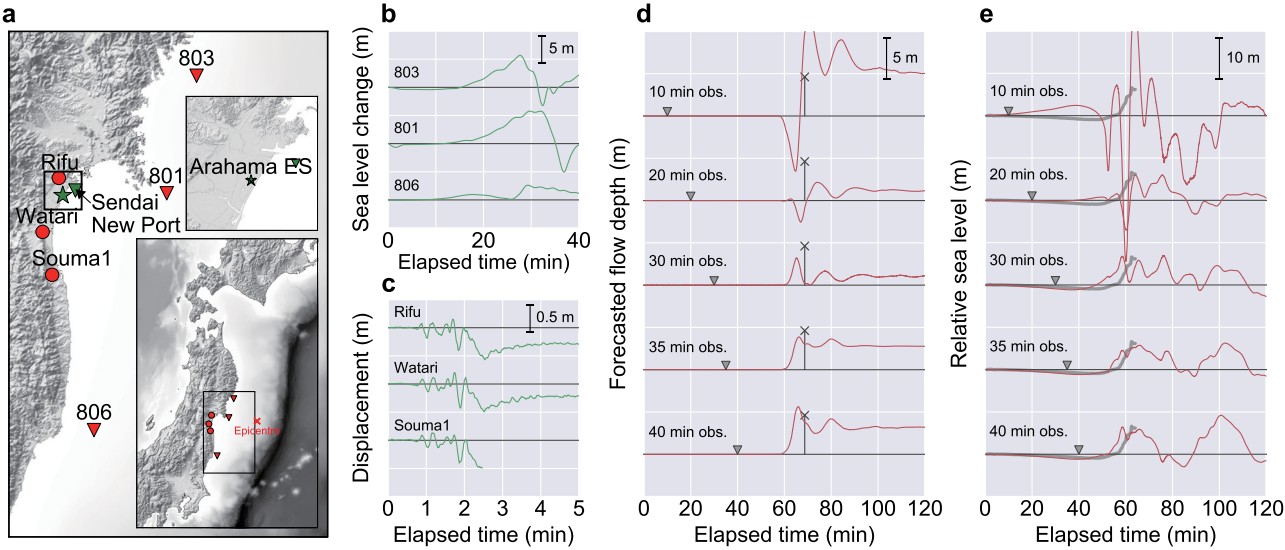

**Fig. 6 Observation points and data for the 2011 Tohoku tsunami for the convolutional neural network (CNN) and forecasting results.** The elapsed time is from the earthquake occurrence. **a** Observation points for the CNN and the forecasting site. The red triangles represent the offshore tsunami observation points (803, 801, 806), and the red circles represent onshore geodetic observation points (Rifu, Watari, Souma1). The red cross mark in the small-scale map represents the epicentre of the 2011 earthquake. The forecasting site is shown in the large-scale map as the green star. Arahama ES (the black cross mark) where tsunami inundation traces for validation are available is located close to the forecasting site. **b** Tsunami waveforms at 803, 801 and 806 observed during the 2011 event[31]. **c** Geodetic observations at Rifu, Watari and Souma1 observed during the 2011 event. **d** Results of tsunami inundation forecasting with different observation periods. Grey triangles represent the observation interval used for forecast. Red waveforms are the CNN forecasting results. The height and position of the black cross mark represents the surveyed tsunami inundation trace at Arahama ES. **e** Results of offshore tsunami waveform forecasting at the Sendai New Port with different observation periods. Grey triangles represent the observation interval used for forecast. Red waveforms are CNN forecasting results. Grey lines are observed waveforms at the Sendai New Port during the 2011 event. Full observation was not available because the gauge was destroyed by the tsunami.

the actual tsunami arrival. A possible cause of this earlier arrival tendency is the effect of rupture propagations, i.e., when generating tsunami data sets for the CNN, the effect of rupture delays on each sub-fault was not considered, and instantaneous slip was assumed; however, the tsunami source inversion assuming sub-faults with multiple time windows suggested that the duration of the tsunamigenic slip of the 2011 Mw 9.0 earthquake lasted ~2.5–3 min[35], and this duration was consistent with the difference in the arrival time. The current CNN forecast provides a slightly earlier arrival tendency and might serve as a cautious warning; however, to forecast the tsunami arrival time more accurately for large earthquakes, it may be necessary to consider the effect of rupture propagation on faults in large earthquakes.

Recent high-sampling-rate tsunami observations, especially by ocean bottom pressure gauges, can capture a wide range of geophysical phenomena in the ocean including ocean currents and seismic waves with much shorter periods (seconds to minutes) than that of tsunamis (minutes to hours)[36]. Consequently, such non-tsunami components can affect tsunami forecasting as noise[37]. To investigate the performance of the CNN under actual observation conditions, we further evaluated the effect of the short-period observation noise on the CNN tsunami forecasting. Noise waveforms were obtained using actual sea-level observation data at the GPS buoys 803, 801 and 806 a day before the 2011 tsunami event (Fig. 7a, b), and the effect of noisy tsunami input on the forecasting results was evaluated using 40 min observation data (Fig. 7c–e). For this evaluation, the similarity between the forecasting results with and without noise were evaluated using the formula for calculating the variance reduction[38]; therefore, the similarity becomes 100% for a perfect match and lower for misfits. The result demonstrated that even with disturbances, the CNN successfully forecasted both the

inundation and offshore waveforms with a small difference. The similarity between forecasting result with and without noise was 99.999% for inundation forecasting and 99.997% for offshore waveform forecasting. Additionally, we also examined the effect of much larger noise on the inundation forecasts. Larger noise waveforms were generated by adding white noise that ranged from −1.0 to 1.0 and was amplified by a certain ratio of the maximum observation amplitude. The additional noise test demonstrated that the CNN tsunami forecast maintained a high similarity of 99.7% on average for 1000 noisy inputs, even with a large noise level of 20% (see Supplementary Fig. 2).

## Discussion

In this study, we presented a tsunami inundation forecasting approach based on a CNN trained with a large quantity of synthetic tsunami scenarios and verified the approach against both synthetic tsunamis and the actual tsunami observations from the 2011 Tohoku earthquake. The CNN tsunami forecast in this study is expected to overcome the bottlenecks of previous simulation-based real-time tsunami forecasting approaches for real applications, such as the difficulty of rapid tsunami source estimations immediately after the earthquake and high computational costs for simulating nonlinear tsunami propagations.

Large earthquakes generating tsunamis are infrequent, resulting in a lack of sufficient data to train the CNN; however, the test using actual observations during the 2011 tsunami event demonstrated that the CNN trained on only synthetic tsunamis can provide an accurate tsunami inundation forecast even for a real tsunami if the target tsunami scenario exists within the distribution of the training data sets. Therefore, it is important to prepare as many tsunami scenarios as possible for training the CNN to address the various mechanisms generating tsunamis.

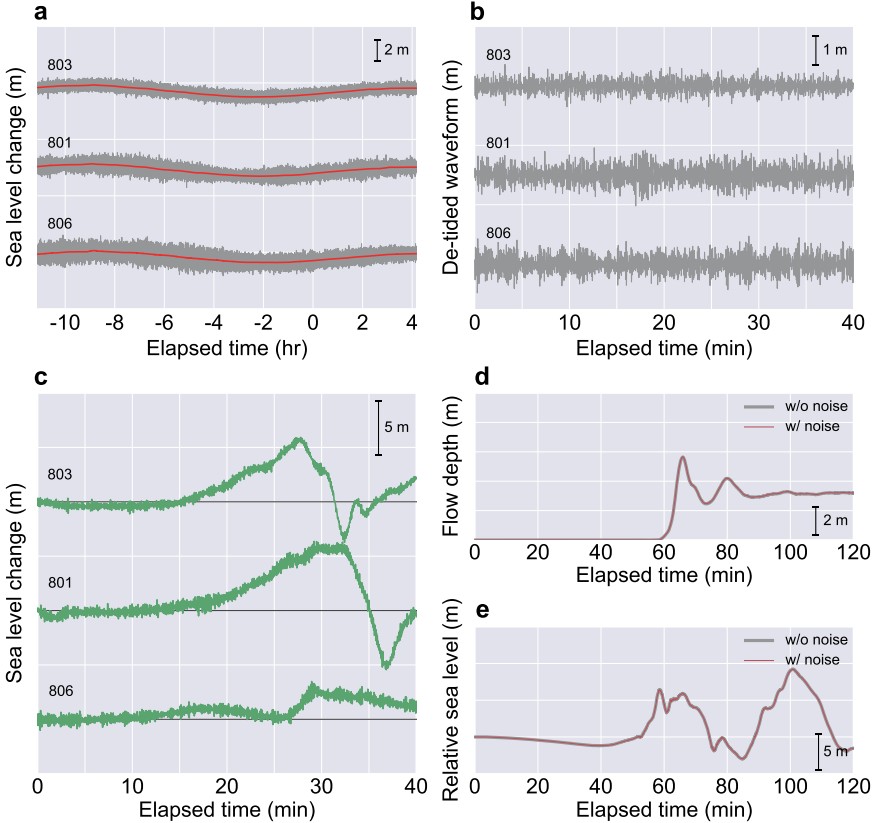

**Fig. 7 Noise waveforms and convolutional neural network (CNN) tsunami forecasting results with and without noise. a** Sea-level observations from GPS buoys before the 2011 tsunami event. The grey and red lines represent the raw observations and the 15 min running average of the raw data, respectively. The elapsed time is from a day before the earthquake occurrence. **b** Noise waveforms extracted from the observations. The deviation from the averaged data (the red lines in **a**) is extracted as noise. The figure shows the noise waveforms during the same observation period for the actual tsunami observations, but a day before the tsunami. **c** Noise waveforms for CNN input. The extracted noise waveforms before the event are added to the actual tsunami observations to prepare the noise inputs. **d** Inundation forecasting results with and without noise. The grey and red lines represent the forecasting results without and with noise, respectively. Because the difference between the two waveforms is small, the result without noise is illustrated with a thicker line. **e** Offshore tsunami forecasting results with and without noise. The grey and red lines represent the forecasting results without and with noise, respectively. Because the difference between the two waveforms is small, the result without noise is illustrated with a thicker line.

Large tsunamis can be caused by large slip at plate boundaries, which is a common tsunami generation mechanism, as well as different types of mechanisms, such as a steep angle slip of splay faults[39], outer-rise normal faults[40] and slow slip at a shallow plate boundary (tsunami earthquake)[41]. Additionally, non-seismic sources, such as volcanic eruptions[42] and landslides[43], can also cause large tsunamis. To address these various tsunami sources by the CNN tsunami forecasting, a wide variety of tsunami scenarios should be included in the training set. A promising solution to address the various types of tsunami is to generate synthetic tsunami scenarios directly assuming a sea surface fluctuation (e.g., using Gaussian distributions having a range of several kilometres or tens of kilometres) rather than considering a sea surface deformation based on fault movements. Local tsunamis caused by volcanic eruptions or landslides can be represented with a few sea surface displacement units[44,45], and the initial tsunami profiles generated by large earthquakes can also be represented by a superposition of a series of sea surface displacement units[46–48]. Simulating a wide variety of tsunami scenarios and training CNNs on large data sets are computationally expensive; however, it is feasible given the recent advances in high-performance computing of tsunami simulations and an efficient training approach as demonstrated in this study. CNN tsunami forecasting trained on various sea surface displacements should have the potential to be applied to a wide variety of tsunamis, including

non-seismic tsunamis, for which the issuance of early warnings has been difficult by employing conventional earthquake-triggered approaches.

## Methods

**Tsunami simulation**. We used the TUNAMI-N2 code[49,50], which is distributed by the Tsunami Inundation Modelling Exchange project of the International Union of Geodesy and Geophysics and Intergovernmental Oceanographic Commission of United Nations Educational, Scientific and Cultural Organisation[50], to create the tsunami simulation data for training the CNNs. The TUNAMI-N2 solves the following nonlinear shallow water Eqs. (1)–(3) with a staggered-grid finite-difference method:

$$\frac{\partial \eta}{\partial t} + \frac{\partial M}{\partial x} + \frac{\partial N}{\partial y} = 0 \tag{1}$$

$$\frac{\partial M}{\partial t} + \frac{\partial}{\partial x}\left(\frac{M^2}{D}\right) + \frac{\partial}{\partial y}\left(\frac{MN}{D}\right) + gD\frac{\partial \eta}{\partial x} + \frac{gn^2 M\sqrt{M^2+N^2}}{D^{7/3}} = 0 \tag{2}$$

$$\frac{\partial N}{\partial t} + \frac{\partial}{\partial x}\left(\frac{MN}{D}\right) + \frac{\partial}{\partial y}\left(\frac{N^2}{D}\right) + gD\frac{\partial \eta}{\partial y} + \frac{gn^2 N\sqrt{M^2+N^2}}{D^{7/3}} = 0 \tag{3}$$

where $\eta$ is the tsunami height, $D$ is the water depth, and $M$ and $N$ are the velocity fluxes in the $x$ and $y$ directions, respectively. $g$ is the gravitational acceleration (=9.81 m/s²), and $n$ is Manning's roughness coefficient, which we set to 0.025 s/m$^{1/3}$ in the simulations in this paper. As employed in general tsunami simulations, nested grid configurations were prepared in which the grid size was decreased by a factor of 3 (1215, 405, 135, 45 and 15 m) to reduce the computational cost; accordingly, tsunamis in coastal areas are evaluated at higher resolutions than those in offshore areas. Bathymetry data projected onto the Japan Plane Rectangular CS X were used for the simulation. The entire calculation domain with nested grids is

illustrated in Supplementary Fig. 3. The finest domain ($\Delta x = \Delta y = 15$ m) covers the Sendai Plain, which experienced a large inundation extent and devastating damage during the 2011 tsunami. The time step of the simulation was set to $\Delta t = 0.2$ s.

**Sensitivity analysis**. To evaluate the sensitivity at each station, we set the signal from a station to zero and evaluated the change in the MSE over 1000 test scenarios. An occlusion test was conducted for every observation point, and the sensitivity of an observation point was evaluated based on the relative change in the MSE from the baseline MSE (no occlusion). The sensitivity is calculated by Eq. (4):

$$\text{Sensitivity} = \frac{\sum_n \text{MSE}' - \sum_n \text{MSE}}{\sum_n \text{MSE}} \qquad (4)$$

where $n$ is the number of test scenarios, $\text{MSE}'$ is the MSE with occlusion, and MSE is the MSE without occlusion (baseline).

## Data availability

The tsunami observation data used in this study are available from The Nationwide Ocean Wave information network for Ports and HArbourS, NOWPHAS (https://www.mlit.go.jp/kowan/nowphas/index_eng.html). The GNSS observation data processed by Shu and Xu are used and available from (https://doi.pangaea.de/10.1594/PANGAEA.914110). Other relevant data in this study are available from the corresponding author upon reasonable request.

## Code availability

The code that supports the findings in this study are available from the corresponding author upon reasonable request.

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

## Acknowledgements
The computational resources of the AI Bridging Cloud Infrastructure (ABCI) provided by the National Institute of Advanced Industrial Science and Technology (AIST) were used. The topography and bathymetry data used in the tsunami numerical simulation were obtained by integrating the data from the Central Disaster Prevention Council; Tohoku Regional Development Bureau of Ministry of Land, Infrastructure, Transport and Tourism (MLIT); and Geospatial Information Authority of Japan. We used the observation data of the 2011 Tohoku tsunami obtained from the Nationwide Ocean Wave information network for Ports and HArbourS, NOWPHAS. The NOWPHAS tsunami and tidal observation data are observed by Ports and Harbours Breau, MLIT and processed by Port and Airport Research Institute.

## Author contributions
F.M. and Y.O. designed the research. F.M. and T.Y. prepared the simulation codes. F.I. prepared the data during the 2011 tsunami event. F.M. and T.F. wrote the manuscript. T.F. and F.I. contributed to data interpretation and provided discussions to improve the quality of the paper.

## Competing interests
The authors declare no competing interests.
