## [Peer Review File · Nature Communications]

REVIEWER COMMENTS

Reviewer #1 (Remarks to the Author):

The manuscript describes results of a study that uses convolutional neural networks approach in place of real-time simulations for tsunami amplitude and arrival forecast. The 2011 Japan Tohoku tsunami fault configuration was used for training and testing scenarios. The paper claims extremely fast forecast speeds and reasonable accuracy for this new approach of the tsunami forecast.

General description and comments:

The study subject is very interesting, it uses a novel approach for fast tsunami forecasting. However, the manuscript leaves many obvious questions unanswered, some possibly due to the brevity of the presentations, but some clearly due to the incompleteness of the study, which, as the result, appears rushed to the publication a bit too soon. More research need to be completed before the claim of the authors about the readiness of the new method for forecasting is substantiated.

Some of the shortcomings and future directions of this study are rightly noted in the discussion session of the manuscript, which is a great 'to do list' for the authors: applying real observation data to assess errors, more training scenarios and the effect on unexpected tsunami sources etc. But yet some more important steps for justifying and testing the method are missing. These steps should be definitely done before publishing a convincing paper. Without these minimum method benchmarks, the manuscript appears very declarative and even speculative at times regarding many claims. Some of the general concerns are outlined bellow, along with some specific comments on the manuscript text.

One obvious test that is missing here is to verify the method with the observation data. Since the training is done on modeling tsunamis originated from the fault section of the 2011 Tohoku tsunami, it is very surprising that the study did not use ample observation data of that tsunami for testing the method. While it wouldn't answer all the questions of applicability, it would establish the very important minimum criteria for the method usability and accuracy when applied to real-world data. There are many offshore observations of this tsunami that can be used for input, and there are plenty of coastal observation that can be used for neural network forecast comparisons. If that test is complimented with the sensitivity analysis with new observation network presented in the current manuscript (albeit improved as outlined bellow), it would present a compelling case for the method usability. Without such test, the paper is simply a sensitivity analysis of the neural network algorithm in application to selected tsunami simulations, and any practical applications of this method are very uncertain.

One more important question for practical application of the method is how the method can forecast unexpected tsunami sources, the ones that are not similar to those used for training the neural network. It appears that the neural network was trained and tested by scenarios from the same set of faults from Figure 5 (it is actually not clear from the text and is my assumption at this point). If this is true, a tsunami from a different type of fault (a splay fault, an outer-rise fault or other mechanisms that have historically happened offshore Japan) may produce quite different forecast result (or not) using the CNN forecast. This problem is alluded to in the discussion but no remedies or solutions are presented. This is quite important question, since we definitely don't know the mechanism of future tsunamis. If we can only forecast mechanisms of past tsunamis, the method is definitely very limited. Test for such situation should be presented or, at the very list, such situation should be thoroughly discussed.

The manuscript is generally well-written, however I've noticed some (certainly not all) grammar and stile issues, so the text may use another thorough grammar check.

Specific comments:

Page 3. The references in this description are limited to just Japan authors, while much bigger literature body on the subject of tsunami studies exist. It may be useful to expand the references here for the sake of readers. Some additional literature that may be included: Line 43: Titov, V.V., A.B. Rabinovich, H.O. Mofjeld, R.E. Thomson, and F.I. González (2005): The global reach of the 26 December 2004 Sumatra Tsunami.. *Science*, 309(5743), 2045–2048, DOI: 10.1126/science.1114576. Line 49: Kânoglu, U., V.V. Titov, E. Bernard, and C. Synolakis (2015): Tsunamis: Bridging science, engineering and society. *Philos.*

Trans. R. Soc. Lond. A, 373(2053), 20140369, doi: 10.1098/rsta.2014.0369. Line 49: Bernard, E., and V.V. Titov (2015): Evolution of tsunami warning systems and products. Philos. Trans. R. Soc. Lond. A, 373(2053), 20140371, doi: 10.1098/rsta.2014.0371. Line 50: Tang, L., V.V. Titov, E. Bernard, Y. Wei, C. Chamberlin, J.C. Newman, H. Mofjeld, D. Arcas, M. Eble, C. Moore, B. Uslu, C. Pells, M.C. Spillane, L.M. Wright, and E. Gica (2012): Direct energy estimation of the 2011 Japan tsunami using deep-ocean pressure measurements. J. Geophys. Res., 117, C08008, doi: 10.1029/2011JC007635.

Line 102: It would be very useful to provide more information on the test sources here, may be a summary figure of sorts. Also, some more information on those 10,000 training scenarios would be useful. It would help to understand the important question: how close the test scenarios are to the training scenarios. The way it is presented, it is not entirely clear.

Line 105: "maximum tsunami amplitude" — Is that at one location, over whole area, over all simulation time?

Line 106: "maximum tsunami arrival time" — please define, it is not clear what does that mean.

Line 106: 0.378 m error doesn't tell much. It may be better to express in relative error, or provide what the maximum amplitudes are. 132 s travel time error sounds like a large error. It is not clear in the following text how it was reduced to 23.9 s. Please explain better.

Line 112: "per tsunami scenario" — it may be better to delete, otherwise it confuses the sentence.

Line 127: replace "long-wave length tsunami" with "tsunami long wave-length"

Line 135: This section and elsewhere — it is not clear if sensitivity is discussed only for one forecast point, or for the whole inundation area. In fact, it is not clearly stated in the manuscript what exactly is forecasted here. Is it inundation area or the amplitude at one point. Is all the training for neural network done for one-point amplitude? If yes, what it would take to train for many points? Please explain more thoroughly in the beginning of the manuscript and in Methods.

Line 158: Replace "rapidity" with "speed"

Line 163: "The results showed a linear increase in the computational cost" — a graph of that linear increase would be useful here, otherwise, this sounds too declarative. The authors invite us to believe. I would avoid such declarations if at all possible.

Line 180: add "in average" to read "only 0.004 s on average" — to be consistent with previous description.

Line 181-182: "While it is often difficult to interpret the results of AI-based prediction methods, the key predictive features revealed by the sensitivity analysis were physically interpretable" This declaration is not clear at all. Please explain more.

Line 197-200: "Additionally, since the CNN predicts tsunami inundation based solely on tsunami waveforms or geodetic observations, the present approach can be used to predict not only earthquake-induced tsunamis but also other types of tsunamis induced by volcanic eruptions or landslides..." This is indeed a very important feature of the method. However, it would require training with models that cover the range of all uncertainties of such sources, wouldn't it? Which may be a very formidable task (or not). The manuscript would definitely use some more discussion on expanding this method to other sources, since this is such an important potential advantage. Otherwise, this is another very declarative sentence with not much explanations or support.

Line 209: The Methods section could be moved to supplement material, if more space is needed for additional tests descriptions, figures and discussions.

Line 232: 12,000 tsunami sources could use some more explanations, may be a summary figure of sorts. Too much of important information is left out of the manuscript description, which does not allow to fully evaluate the results here.

Line 241: In this section, it may be useful to provide information on how expensive the training is in terms of computational resources and time.

Figure 1: Please define the blue star in the caption (I assume it is forecast point where all results are discussed — it is not clear from the whole manuscript)

Figure 2: Are b and c for the star location? Again, define star.

Reviewer #2 (Remarks to the Author):

The paper proposes a method to predict the evolution of tsunamis based on simulated data from a set of virtual observations with a convolutional neural network. Overall, the deep learning methodology makes sense, and the results indicate a good accuracy for the synthetic data sets used in this study.

While some smaller details are missing in the text, I think the eposition should be revised in order to clarify what the paper has achieved. If I understood it correctly, only synthetic data was used and evaluated. This is an interesting step by itself, but does not allow for conclusions being draw about accuracy or reliability in real world settings. In a way, this only allows for clear statements about the runtime performance, which is indeed impressive. However, as it is unclear whether the training methodology carries over to real world data, this is only a partial step toward enabling deep-learning based tsunami forecasting.

Hence, I think a very important update for a revised version will be to clarify early on that the present study focuses on synthetic data, and hence does not allow for conclusions being drawn about real world accuracies. Right now parts of the text (e.g., L28 of the abstract, and L68 of the intro) are potentially misleading. In L48, real-world observations are mentioned, and I was expecting an evaluation to use these later on. Again, it would be important to clarify that the present study does not use real-world data, but instead uses a custom, simulated data set. The preliminary aspects are mentioned in lines L188 and 204, which unfortunately is too late. It is important to clarify this early on in the introduction and abstract. Alternatively, as the authors mention the real-world data, maybe they could include results evaluating real-world data? This would require a larger update, but certainly strengthen the paper. If this is not possible, textual clarifications regarding the use of synthetic data as mentioned in this paragraph should be added.

Looking ahead toward the discussion section, I would also recommend to add a paragraph discussing the general availability of real-world data, and to which extent the proposed methodology could be applied with real-world data. (E.g. is there enough data to train a model with more than 10000 real world cases?)

The CNN architecture makes sense, and most of the details are given in the text and the appendix. A stack of 1D convolutions with an additional tail of fully connected layers is used. The data set with 12000 simulations, 1000 of which were used for testing sounds very thorough. One aspect that I couldn't find in the text was the temporal size of the output window. I.e., how many eta values over time are predicted by the last FC layer? Related to this, did I understand it correctly that only a single spatial location is used? Wouldn't it be relatively easy to evaluate different locations, or were the simulations performed specifically for the Tohoku location?

I have to admit I also didn't fully understand the structure of the inputs. What does a 5 minute segment of observation data encompass, i.e., how many values in space and time? Table 1 indicates that the network size grows by 13 million for each 5 minutes, but the number of channels seem to be fixed. So are these 13m weights really added in the first convolutional layer? This seems a bit unbalanced, but I'm not sure I understood this part of the NN architecture correctly.

Regarding L388, can the authors clarify: are they fully committed to releasing code and data, i.e. making it freely available, or do they plan to only selectively allow access?

Overall, I think with tsunami forecasting this work targets a very important direction. As such, I think it's very interesting to see advances via deep-learning in this area. Before publication, I think it's important to clarify the points above within a revision, hence the focus on synthetic data, and the unclear points regarding the NN architecture.

Reviewer #3 (Remarks to the Author):

Dear authors of "Early Forecasting of Tsunami Inundation from Tsunami and Geodetic Observation Data with Convolutional Neural Networks".

I'd like to congratulate you on the work you have done for this paper - this is an interesting study that shows how CNN's can be used for rapid tsunami forecasting. The paper illustrates nicely how CNNs can be used for this task and how it would speed up the forecasting time drastically with acceptable accuracy.

I think the more technical parts of the paper are written well, but the 'Results' section needs more work. Overall I'd like to ask you to replace qualitative statements as much as feasible with quantitative statements, e.g. "much bigger" with "twice as much" etc.

Also please describe better and earlier in the text what the data basis for training, validation and testing actually consists of (how many scenarios, what source region, how many magnitudes considered, non-uniform slip used ? how was it generated ...).

Another point to me is that it is difficult to understand what the CNN actually predicts: Is it one single time series at one location on land ? Is it the whole inundation grid with time development? Please describe this in more detail.

Can you also please discuss more explicitly where the method would not work (distant source tsunami ?). You state it should work for landslide tsunami sources. These statements should be phrased more carefully and the limitations should be discussed.

Please also consider my comments in the annotated MS, which point out these issues at specific points in the text.

I don't think the research needs more work, but the text does.

I recommend moderate revision.

Responses to reviewers

We would like to express our sincere thanks to the reviewers for their valuable and insightful comments and suggestions. All of their comments were helpful for improving the paper, and we addressed all the comments as shown in the following responses. In the revised manuscript, we additionally investigated the applicability of the proposed method to actual tsunamis using real-world data obtained during the 2011 Tohoku earthquake. To be consistent with new figures for additional results, unit of the elapsed time in the original figures was also modified.

In the following responses, the reviewers' comments are coloured in black, and our point-by-point responses are positioned right after the comments and coloured in blue. The modified parts in the revised manuscript are highlighted.

Reviewer #1 (Remarks to the Author):

The manuscript describes results of a study that uses convolutional neural networks approach in place of real-time simulations for tsunami amplitude and arrival forecast. The 2011 Japan Tohoku tsunami fault configuration was used for training and testing scenarios. The paper claims extremely fast forecast speeds and reasonable accuracy for this new approach of the tsunami forecast.

> We again greatly appreciate your valuable comments on our paper. All of your comments were constructive and effective to improve the paper. For the raised concerns, especially about the applicability to real events using actual data, we conducted additional experiments and incorporated into the revise the manuscript.

General description and comments:

The study subject is very interesting, it uses a novel approach for fast tsunami forecasting. However, the manuscript leaves many obvious questions unanswered, some possibly due to the brevity of the presentations, but some clearly due to the incompleteness of the study, which, as the result, appears rushed to the publication a bit too soon. More research need to be completed before the claim of the authors about the readiness of the new method for forecasting is substantiated.

Some of the shortcomings and future directions of this study are rightly noted in the

discussion session of the manuscript, which is a great 'to do list' for the authors: applying real observation data to assess errors, more training scenarios and the effect on unexpected tsunami sources etc. But yet some more important steps for justifying and testing the method are missing. These steps should be definitely done before publishing a convincing paper. Without these minimum method benchmarks, the manuscript appears very declarative and even speculative at times regarding many claims. Some of the general concerns are outlined below, along with some specific comments on the manuscript text.

> As you pointed out, the original manuscript lacked sufficient results to support our claim. In the revised manuscript, we investigated the performance of the CNN using real-world observation obtained during the 2011 Tohoku tsunami event and reported the results. We also strengthened discussions on the applicability of the proposed method to unexpected tsunamis.

One obvious test that is missing here is to verify the method with the observation data. Since the training is done on modeling tsunamis originated from the fault section of the 2011 Tohoku tsunami, it is very surprising that the study did not use ample observation data of that tsunami for testing the method. While it wouldn't answer all the questions of applicability, it would establish the very important minimum criteria for the method usability and accuracy when applied to real-world data. There are many offshore observations of this tsunami that can be used for input, and there are plenty of coastal observation that can be used for neural network forecast comparisons. If that test is complimented with the sensitivity analysis with new observation network presented in the current manuscript (albeit improved as outlined below), it would present a compelling case for the method usability. Without such test, the paper is simply a sensitivity analysis of the neural network algorithm in application to selected tsunami simulations, and any practical applications of this method are very uncertain.

> Following your comment and advice, we conducted the forecasting performance test by using actual observation data obtained during the 2011 Tohoku earthquake. After training the CNN on 10,000 synthetic tsunamis with the observation settings during the 2011 Tohoku tsunami event, we examined the forecasting performance by inputting actual observation data of the 2011 Tohoku earthquake. The results confirmed that the CNN could estimate the tsunami inundation from the real data, and the results were consistent with the observed tsunami trace near the forecasting point. In addition, the CNN forecast was validated against using a nearshore tsunami timeseries observation by applying the CNN nearshore tsunami

waveform forecasting.

Moreover, we investigated the noise tolerance of the CNN tsunami forecasting method assuming the actual conditions of tsunami observation. The noise waveforms obtained from actual observations during normal time were added to the actual tsunami observation data of the 2011 event, and we evaluated the forecasting results by applying the trained CNN with noisy inputs. As a result, there was almost no change in the forecasted waveforms even with actual noise. Additional test using larger random noises confirmed that the forecasting result was contaminated with a large noise; however, it demonstrated high noise tolerance of the CNN tsunami forecasting. These results are presented in a new subsection in the Results (Lines 226 – 295 and new Figure 6,7 and Supplementary Figure 2), and Abstract is also revised (Lines 27 – 29).

One more important question for practical application of the method is how the method can forecast unexpected tsunami sources, the ones that are not similar to those used for training the neural network. It appears that the neural network was trained and tested by scenarios from the same set of faults from Figure 5 (it is actually not clear from the text and is my assumption at this point). If this is true, a tsunami from a different type of fault (a splay fault, an outer-rise fault or other mechanisms that have historically happened offshore Japan) may produce quite different forecast result (or not) using the CNN forecast. This problem is alluded to in the discussion but no remedies or solutions are presented. This is quite important question, since we definitely don't know the mechanism of future tsunamis. If we can only forecast mechanisms of past tsunamis, the method is definitely very limited. Test for such situation should be presented or, at the very list, such situation should be thoroughly discussed.

> Since the CNN forecasts tsunamis based on training data sets, high accuracy for future tsunamis cannot be expected if the source and mechanism of tsunamis are not included within the training distribution. Therefore, comprehensive tsunami scenarios should be prepared in the training data for precise tsunami forecast. In addition to ordinary tsunami scenarios, there are different types of fault mechanism that generate tsunamis, such as a splay fault, an outer-rise fault, and a tsunami earthquake. Additionally, non-seismic sources such as volcanic eruptions and landslides can also cause or amplify tsunamis. A potential solution to address these unexpected tsunami sources is the creation of various tsunami scenarios using multiple unit sources of sea surface changes rather than assuming fault scenarios. A unit of sea surface change can represent tsunamis caused not only by earthquakes using multiple units (e.g., Tsushima et al., 2011; Saito et al., 2011; Hossen et al.,

2015) but also by volcanic eruptions (e.g., Fukao et al., 2018) and landslides (e.g., Watts et al., 2005) using fewer units. We consider that the CNN trained on such a variety of tsunami scenarios has the potential to forecast a wide variety of tsunamis. In the revised manuscript we discussed in detail the limitation of the CNN forecast for unexpected sources and the potential solution to address the unexpected tsunamis (Lines 305 – 328).

References:

Saito, T., Ito, Y., Inazu, D. & Hino, R. Tsunami source of the 2011 Tohoku - Oki earthquake, Japan: Inversion analysis based on dispersive tsunami simulations. *Geophysical Research Letters* 38, L00G19 (2011).

Tsushima, H. et al. Near-field tsunami forecasting using offshore tsunami data from the 2011 off the Pacific coast of Tohoku Earthquake. *Earth, Planets and Space* 63, 56 (2011).

Hossen, M.J., Cummins, P.R., Dettmer, J. & Baba, T. Tsunami waveform inversion for sea surface displacement following the 2011 Tohoku earthquake: Importance of dispersion and source kinematics. *J. Geophys. Res. Solid Earth* 120, 6452–6473 (2015).

Fukao Y. et al. Mechanism of the 2015 volcanic tsunami earthquake near Torishima, Japan. *Science Advances* 4, eaao0219 (2018).

Watts, P., Grilli S.T., Tappin, D.R. & Fryer G.J. Tsunami Generation by Submarine Mass Failure. II: Predictive Equations and Case Studies. *Journal of Waterway, Port, Coastal, and Ocean Engineering* 131, 298–310 (2005).

The manuscript is generally well-written, however I've noticed some (certainly not all) grammar and style issues, so the text may use another thorough grammar check.

> Thank you for your careful reading. The revised manuscript has taken your suggestions and has been processed by an editing service.

Specific comments:

Page 3. The references in this description are limited to just Japan authors, while much bigger literature body on the subject of tsunami studies exist. It may be useful to expand the references here for the sake of readers. Some additional literature that may be included:

Line 43: Titov, V.V., A.B. Rabinovich, H.O. Mofjeld, R.E. Thomson, and F.I. González (2005): The global reach of the 26 December 2004 Sumatra Tsunami. *Science*, 309(5743), 2045–2048, DOI: 10.1126/science.1114576. Line 49: Kânoglu, U., V.V. Titov, E. Bernard, and C. Synolakis (2015): Tsunamis: Bridging science, engineering and society. *Philos. Trans. R. Soc. Lond. A*, 373(2053), 20140369, doi: 10.1098/rsta.2014.0369. Line 49: Bernard, E., and

V.V. Titov (2015): Evolution of tsunami warning systems and products. *Philos. Trans. R. Soc. Lond. A*, 373(2053), 20140371, doi: 10.1098/rsta.2014.0371. Line 50: Tang, L., V.V. Titov, E. Bernard, Y. Wei, C. Chamberlin, J.C. Newman, H. Mofjeld, D. Arcas, M. Eble, C. Moore, B. Uslu, C. Pells, M.C. Spillane, L.M. Wright, and E. Gica (2012): Direct energy estimation of the 2011 Japan tsunami using deep-ocean pressure measurements. *J. Geophys. Res.*, 117, C08008, doi: 10.1029/2011JC007635.

> Following your suggestion, we have revised the citations and incorporated the suggested references into the revised manuscript (Lines 44 – 47 and 50 – 51).

Line 102: It would be very useful to provide more information on the test sources here, may be a summary figure of sorts. Also, some more information on those 10,000 training scenarios would be useful. It would help to understand the important question: how close the test scenarios are to the training scenarios. The way it is presented, it is not entirely clear.

> Based on your comment and comments from other reviewers, we have created a subsection in the earlier part of the manuscript describing the data preparation scheme for the CNN. A summary figure has also been prepared for clarity. Please see Lines 105 – 127 and Figure 2.

Line 105: “maximum tsunami amplitude” — Is that at one location, over whole area, over all simulation time?

> Our CNN forecasts a tsunami inundation waveform at a single onshore point. The “maximum tsunami amplitude” is the maximum value of the forecasted waveform. We have revised the sentences for clarity (Lines 85 – 86 and 157 – 158).

Line 106: “maximum tsunami arrival time” — please define, it is not clear what does that mean.

Line 106: 0.378 m error doesn't tell much. It may be better to express in relative error, or provide what the maximum amplitudes are. 132 s travel time error sounds like a large error. It is not clear in the following text how it was reduced to 23.9 s. Please explain better.

> In the revised manuscript, the error metrics are clearly defined, and errors are also presented in relative errors according to your comment. In the original manuscript, we used

two terms regarding the timing of tsunami: “maximum tsunami arrival time” (the arrival time of the maximum tsunami amplitude) and “tsunami arrival time” (the timing of initiation of tsunami inundation defined as the time when the tsunami flow depth first exceeds 10% of the maximum flow depth). Based on your comment, we aware that the original metrics were misleading. Additionally, the “maximum tsunami arrival time”, simply picking up the time of the maximum tsunami, is not suitable for evaluating tsunami inundation since the same flow depths can be observed after the arrival of tsunami. Thus, in the revised manuscript, we used only “tsunami arrival time” for evaluating tsunami inundation forecasting to avoid confusion. During the revision, we became aware of an error estimation of the tsunami arrival time and corrected it in the revised manuscript. We apologize for this error in the original manuscript and confusion associated with the error. Please see Lines 156 – 163.

Line 112: “per tsunami scenario” — it may be better to delete, otherwise it confuses the sentence.

> We deleted the sentences according to your suggestion. Please see Lines 163 – 164.

Line 127: replace “long-wave length tsunami” with “tsunami long wave-length”

> We replaced “long-wave length tsunami” with “tsunami long wave-length”. Please see Line 178.

Line 135: This section and elsewhere — it is not clear if sensitivity is discussed only for one forecast point, or for the whole inundation area. In fact, it is not clearly stated in the manuscript what exactly is forecasted here. Is it inundation area or the amplitude at one point. Is all the training for neural network done for one-point amplitude? If yes, what it would take to train for many points? Please explain more thoroughly in the beginning of the manuscript and in Methods.

> The CNN in this paper forecasts the time series of tsunami inundation waveform at a single location (green star in Figure 1). It is possible to forecast many points by preparing multiple networks. We added this explanation for clarity. Please see Lines 85 – 86 and 102 – 103.

Line 158: Replace “rapidity” with “speed”

> We replaced this word as suggested. Please see Line 209.

Line 163: “The results showed a linear increase in the computational cost” — a graph of that linear increase would be useful here, otherwise, this sounds too declarative. The authors invite us to believe. I would avoid such declarations if at all possible.

> As you pointed out our claim for linearity is declarative. We have removed the term “linear” from the revised manuscript. Due to limited space, we did not include the figure. Please see Line 211 – 214.

Line 180: add “in average” to read “only 0.004 s on average” — to be consistent with previous description.

> According to the revisions and additional results using real-world data, the Discussion section was revised, and the corresponding part was removed. Please see Lines 298 – 304.

Line 181-182: “While it is often difficult to interpret the results of AI-based prediction methods, the key predictive features revealed by the sensitivity analysis were physically interpretable” This declaration is not clear at all. Please explain more.

> Because of the additional result for the 2011 tsunami event and the limited space, the Discussion section was revised, and the corresponding part was removed in the revised manuscript. What we meant here in the original manuscript is explained in the revised previous sections (Lines 202 – 205).

Line 197-200: “Additionally, since the CNN predicts tsunami inundation based solely on tsunami waveforms or geodetic observations, the present approach can be used to predict not only earthquake-induced tsunamis but also other types of tsunamis induced by volcanic eruptions or landslides...” This is indeed a very important feature of the method. However, it would require training with models that cover the range of all uncertainties of such sources, wouldn't it? Which may be a very formidable task (or not). The manuscript would definitely use some more discussion on expanding this method to other sources, since this is such an important potential advantage. Otherwise, this is another very declarative sentence with not much explanations or support.

> Related to the previous comments and reply to the CNN forecasts of unexpected tsunami,

we have added a discussion about the applicability of the proposed method to non-earthquake-induced tsunamis with references (please see Lines 311 – 322). Based on other comments, we also added information about the required time to train the CNN model (training could be completed within 2 hours using several GPUs; see Lines 150 – 151). Given the training time and recent advances in high performance computing of tsunami simulations, we believe that it is possible to prepare large tsunami simulation data sets and built more generalized CNN models. We have added this discussion to the revised manuscript. Please see Lines 322 – 325.

Line 209: The Methods section could be moved to supplement material, if more space is needed for additional tests descriptions, figures and discussions.

> Thank you for your kind suggestion. To create space for content on the new tests with real data, we moved Figure 5 and Table 2 to the supplementary material.

Line 232: 12,000 tsunami sources could use some more explanations, may be a summary figure of sorts. Too much of important information is left out of the manuscript description, which does not allow to fully evaluate the results here.

> According to your comment and comments from other reviewers, we have created a subsection in the earlier part of the revised manuscript describing the data preparation scheme for the CNN with a summary figure. We have also created a subsection describing the training process just after the data section. We think the readers will now be able to follow the procedure for CNN tsunami forecasting: framework, data preparation, training and forecasting. Please see Lines 105 – 127 and Figure 2.

Line 241: In this section, it may be useful to provide information on how expensive the training is in terms of computational resources and time.

> We described the time required for training the CNN models. This depends on the network size, but even the largest network in this study can be completed within 2 hours. We added a sentence regarding the training cost. Please see Lines 150 – 151.

Figure 1: Please define the blue star in the caption (I assume it is forecast point where all results are discussed — it is not clear from the whole manuscript)

> We defined what the star represents in the revised figures and manuscript. Please see Lines 85 – 86, 100 – 101 and the revised caption for Fig.1 (Lines 528 – 530).

Figure 2: Are b and c for the star location? Again, define star.

> Yes. The results presented in the original Figure 2 b and c are for the location of the star, where the CNN predicts the tsunami inundation waveform. We defined the star in the revised sentences and figures for clarity. Please see the revised caption for the figure (Lines 545 – 546).

Reviewer #2 (Remarks to the Author):

The paper proposes a method to predict the evolution of tsunamis based on simulated data from a set of virtual observations with a convolutional neural network. Overall, the deep learning methodology makes sense, and the results indicate a good accuracy for the synthetic data sets used in this study.

> Thank you for your constructive comments and understanding of our work. We have revised the manuscript by adding the results of the CNN performance survey using real data from the 2011 Tohoku earthquake. We further investigated the performance of the CNN with noisy observation inputs, considering real-world situations.

While some smaller details are missing in the text, I think the eposition should be revised in order to clarify what the paper has achieved. If I understood it correctly, only synthetic data was used and evaluated. This is an interesting step by itself, but does not allow for conclusions being draw about accuracy or reliability in real world settings. In a way, this only allows for clear statements about the runtime performance, which is indeed impressive. However, as it is unclear whether the training methodology carries over to real world data, this is only a partial step toward enabling deep-learning based tsunami forecasting.

Hence, I think a very important update for a revised version will be to clarify early on that the present study focuses on synthetic data, and hence does not allow for conclusions being drawn about real world accuracies. Right now parts of the text (e.g., L28 of the abstract, and L68 of the intro) are potentially misleading. In L48, real-world observations are mentioned, and I was expecting an evaluation to use these later on. Again, it would be important to clarify that the present study does not use real-world data, but instead uses a custom,

simulated data set. The preliminary aspects are mentioned in lines L188 and 204, which unfortunately is too late. It is important to clarify this early on in the introduction and abstract. Alternatively, as the authors mention the real-world data, maybe they could include results evaluating real-world data? This would require a larger update, but certainly strengthen the paper. If this is not possible, textual clarifications regarding the use of synthetic data as mentioned in this paragraph should be added.

> In response to your comments and concerns from other reviewers, we further investigated CNN's performance against actual observation data from the 2011 Tohoku tsunami event and added the results to the revised manuscript. We newly trained the CNN on the same 10,000 simulated synthetic tsunamis used in the original manuscript under the real-world observation settings in 2011 Tohoku tsunami event. We then investigated the performance of the CNN by inputting actual observations from the 2011 event into the CNN and comparing the forecasting results with surveyed tsunami traces near the forecasting point. After obtaining sufficient observations, the CNN successfully forecasted the actually observed tsunami height and its arrival time. Additionally, we created noisy tsunami observation using noise obtained from actual sea level observations during normal time, and we confirmed the effect of noise on the forecasting performance. The test results demonstrated that the CNN has a robust noise tolerance. We believe that these additional results strengthen our paper and support our claim made in the original manuscript. Please see Lines 226 – 295 and new Figure 6, 7 and Supplementary Figure 2.

Looking ahead toward the discussion section, I would also recommend to add a paragraph discussing the general availability of real-world data, and to which extent the proposed methodology could be applied with real-world data. (E.g. is there enough data to train a model with more than 10000 real world cases?)

> Application of the CNN forecast to real-world data from the 2011 event was added in the revised manuscript (Lines 226 – 295). Forecast for unexpected tsunamis such as caused by volcanic eruptions and landslides which are not currently used for CNN training is discussed in the Discussed section. Please see Lines 305 – 328.

The CNN architecture makes sense, and most of the details are given in the text and the appendix. A stack of 1D convolutions with an additional tail of fully connected layers is used. The data set with 12000 simulations, 1000 of which were used for testing sounds very thorough. One aspect that I couldn't find in the text was the temporal size of the output

window. I.e., how many eta values over time are predicted by the last FC layer? Related to this, did I understand it correctly that only a single spatial location is used? Wouldn't it be relatively easy to evaluate different locations, or were the simulations performed specifically for the Tohoku location?

> We apologize for the insufficient description. The CNN predicts an eta value with a length of 3600, as described for the FC3 layer in the original Table 2. This is equivalent to forecasting for approximately 7200 s since the predicted eta value has a sampling frequency of 0.5 Hz. The CNN predicts a time series of eta values at a single point (green star in Figure 1). In this paper, we conducted tsunami simulations specifically for the Tohoku region since there is an observation network that can be input for the CNN, and rich actual observations and surveys from the 2011 event are available to investigate the performance of the CNN using real-world data. Of course, the presented framework can be applied to different areas under conditions of various station arrangements and earthquake-tsunami generation environments. Based on your comments, we added some sentences to clarify this in the revised manuscript. Please see Lines 100 – 101 and 117 – 118. A description regarding forecasts at multiple points has also been added. Please see Lines 102 – 103.

I have to admit I also didn't fully understand the structure of the inputs. What does a 5 minute segment of observation data encompass, i.e., how many values in space and time? Table 1 indicates that the network size grows by 13 million for each 5 minutes, but the number of channels seem to be fixed. So are these 13m weights really added in the first convolutional layer? This seems a bit unbalanced, but I'm not sure I understood this part of the NN architecture correctly.

> The increased parameters are not for the first layer, but mainly due to the larger size of the fully connected layer after the convolutional layers. The inputs for the CNN were data with a 1 Hz sampling and thus having different lengths according to the observation time; The array length for 5-minutes observation is 300, and the array length of 30 min observation is 1800. As described in the original Table 2 and the Methods section, the same network configuration was used for different observation lengths to determine the effect of the observation length, so the length of data passing through the network and the number of parameters also increased as increasing the observation length. For example, the number of elements after passing through the conv3_4 layer was approximately 26M for 10 min observations and approximately 79M for 30 min observations. Thus, under the network configuration of this study, the number of required parameters increases as the observation

time increases. We added some explanations to the revised manuscript for clarity. Please see Lines 211 – 216.

Regarding L388, can the authors clarify: are they fully committed to releasing code and data, i.e. making it freely available, or do they plan to only selectively allow access?

> Since we have provided all relevant information for the implementation (e.g., network configuration and parameters) in the manuscript, we will provide code and data upon request as clarified in the Code and Data availability section. Please see Lines 362 – 371.

Overall, I think with tsunami forecasting this work targets a very important direction. As such, I think it's very interesting to see advances via deep-learning in this area. Before publication, I think it's important to clarify the points above within a revision, hence the focus on synthetic data, and the unclear points regarding the NN architecture.

> Thank you very much again for your valuable and constructive comments. We added additional results with real-world data in addition to synthetic data according to your comments and comments from the other reviewers. We believe that these revisions strengthened our paper and supports our claim.

Reviewer #3 (Remarks to the Author):

Dear authors of "Early Forecasting of Tsunami Inundation from Tsunami and Geodetic Observation Data with Convolutional Neural Networks".

I'd like to congratulate you on the work you have done for this paper - this is an interesting study the shows how CNN's can be used for rapid tsunami forecasting. The paper illustrates nicely how CNNs can be used for this task and how it would speed up the forecasting time drastically with acceptable accuracy.

> Thank you very much for your supportive and constructive comments. Based on your comments and the comments from the other reviewers, we have revised the manuscript.

I think the more technical parts of the paper are written well, but the 'Results' section needs more work. Overall I'd like to ask you to replace qualitative statements as much as feasible with quantitative statements, e.g. "much bigger" with "twice as much" etc.

> We replaced qualitative statements with quantitative statements in the revised manuscript.

Also please describe better and earlier in the text what the data basis for training, validation and testing actually consists of (how many scenarios, what source region, how many magnitudes considered, non-uniform slip used ? how was it generated ...).

> Thank you very much for your suggestion. We moved the contents that were originally presented in the Methods section to an earlier part in the revised manuscript. More detailed information was also added for clarity. Please see Lines 105 – 127.

Another point to me is that it is difficult to understand what the CNN actually predicts: Is it one single time series at one location on land? Is it the whole inundation grid with time development? Please describe this in more detail.

> We revised the manuscript to clarify what the CNN predicts. The caption for Figure 1 was also modified for clarity. Please see Lines 85 – 86, 100 – 101 and 528 – 530.

Can you also please discuss more explicitly where the method would not work (distant source tsunami?). You state it should work for landslide tsunami sources. These statements should be phrased more carefully and the limitations should be discussed.

> In the revised manuscript, we added discussions about the limitations of the proposed method and potential further research directions to overcome those limitations with references. Please see Lines 311 – 328.

Please also consider my comments in the annotated MS, which point out these issues at specific points in the text.

> Thank you very much for carefully reading the manuscript. We addressed all the annotated comments in the revised manuscript.

I don't think the research needs more work, but the text does.

I recommend moderate revision.

> Again, we greatly appreciate your valuable comments to improve our paper.

REVIEWERS' COMMENTS

Reviewer #1 (Remarks to the Author):

The revised manuscript provides much improved description of the method, much more convincing verification of the method accuracy and overall a much better paper with an impressive potential new direction for the development of the tsunami forecast. I recommend publication with some minor adjustments suggested in my specific comments.

Specific comments

Line 124 and Figure 2: I agree with providing the seismic moment as the main earthquake metric, but would also suggest to list corresponding Moment Magnitude (M_w) in addition. It is still a widely used and easily-recognized value.

Line 241: Replace "devastated" with "destroyed".

Line 296: Please indicate units for the large noise. I assume amplitudes, but not sure. The noise sensitivity test is a great addition to the paper. For a more complete test a lower frequency noise may be needed also. The authors may or may not be able to additional comparison to the paper, but I would at least recommend to add a brief discussion on the frequency components of potential noise that would include north the seismic and tsunami waves frequencies.

Line 341-342: The equation numbering is very confusing, it should be separated from the equations. Please fix for the print version.

Figure 6 caption, lines 567-568: "large-scale map" and "small-scale map" have actually the opposite meanings to the ones used here. Large-scale map shows a smaller area with higher resolution and inverse. Please interchange.

Figure 6 caption, lines 572, 576: I would replace "lead-time" with more descriptive "interval used for forecast".

Reviewer #2 (Remarks to the Author):

This is a review of the first revision of an NCOMMS submission that targets predictions of the evolution of tsunamis based on simulated data from a set of virtual observations with a convolutional neural network.

In the revised version, the authors have, most importantly, included an evaluation with real-world observations obtained during the 2011 Tohoku tsunami. This is a meaningful and important addition and helps to make a case for the proposed method. Especially figure 6 is interesting to see. The forecast (naturally) doesn't match the observations exactly, but there's a clear trend and a relatively small offset in the prediction versus the measurement. Additionally, the effect of noise is evaluated with this scenario.

Other improvements include more specifics on error metrics, a discussion of non-earthquake tsunamis, a clarification of the NN architecture and the use of synthetic data (among others).

Overall, I think this version addresses my concerns and I think the manuscript is ready for publication. Especially the addition of the real-world case was good to see and improves the submission substantially.

Reviewer #3 (Remarks to the Author):

I feel the authors have improved the ms a lot in this latest revision. Adding the real world example and assessment is very helpful in putting the method and it's performance into context. It seems inevitable that deep learning will become a part of the warning toolkit in natural hazards and tsunami in particular. This study is important as it shows the potential and the difficulty of the approach. It is novel in its particular application.

My initial comments in the first review have been addressed to my satisfaction and the paper reads much better now and is better to understand.

I only have a few suggestions for minor revision. Please consider the following suggested changes:

L 74: '... predict the tsunami inundation time series at a single location. ...'

L 118-120: Here it is not quite clear what 'length' refers to. I am guessing it is the number of samples that are fed into the CNN. The author should say 'number of samples' instead of 'length' (a length is a measure of distance and would require a unit. Here the authors only give a 'number': 3600.

L 266: '... 20 min or less observation time when ...

L 313: ... to prepare as many [remove: as possible] tsunami scenarios as possible for Training ...

L 324: insert the statement in brackets '(Gaussian distribution ...)' [HERE] '... fluctuation [HERE] rather than ...' (L:322)

Responses to reviewers

We again would like to express our sincere thanks to the reviewers for their careful reading of the manuscript, valuable comments, and suggestions.

In the following responses, we provide our point-by-point responses right after the comments and coloured in blue. In addition to the revisions based on comments and suggestions from reviewers, there are slight modifications by self-review based on editorial instructions. All modified parts in the revised manuscript are highlighted.

Reviewer #1 (Remarks to the Author):

The revised manuscript provides much improved description of the method, much more convincing verification of the method accuracy and overall a much better paper with an impressive potential new direction for the development of the tsunami forecast. I recommend publication with some minor adjustments suggested in my specific comments.

> We again sincerely appreciate your careful reading of the manuscript and positive feedback on the revised manuscript. We addressed all of your comments in the revised manuscript as shown below.

Specific comments

Line 124 and Figure 2: I agree with providing the seismic moment as the main earthquake metric, but would also suggest to list corresponding Moment Magnitude (M_w) in addition. It is still a widely used and easily-recognized value.

> We agree with your opinion. In the figure, we found that converting the seismic moment to M_w on a logarithmic scale resulted in lower visibility of the difference among the scenarios, so that we maintained the figure as displaying seismic moment. Instead, in the text, we have added the corresponding M_w for better understanding. The corresponding M_w range is from 9.0 to 9.2. (Line 126 – 127)

Line 241: Replace “devastated” with “destroyed”.

> The word is replaced as suggested. (Line 243)

Line 296: Please indicate units for the large noise. I assume amplitudes, but not sure. The noise sensitivity test is a great addition to the paper. For a more complete test a lower frequency noise may be needed also. The authors may or may not be able to additional comparison to the paper, but I would at least recommend to add a brief discussion on the frequency components of potential noise that would include north the seismic and tsunami waves frequencies.

> For clarity, we modified the term “signal” to “amplitude” (Line 302). Following your comment, we have added a brief description of the frequency components of potential noise containing seismic waves (Line 285 – 290). Because the predominant frequency of such noise tends to be higher than that of tsunamis, here we focused on the effect of higher frequency noise on the forecasting, rather than lower frequencies noise than that of tsunamis. We have cited two new references (Kubota et al., 2020; Saito and Tsushima, 2016) to explain the characteristics of high-frequency noise in tsunami observations and its potential effects on tsunami forecasting.

Line 341-342: The equation numbering is very confusing, it should be separated from the equations. Please fix for the print version.

> Sorry for any inconvenience associated with the equation numberings, which seems to be occurred in automated online processing. The revised manuscript displays the numbers as intended. (Line 346 – 347)

Figure 6 caption, lines 567-568: “large-scale map” and “small-scale map” have actually the opposite meanings to the ones used here. Large-scale map shows a smaller area with higher resolution and inverse. Please interchange.

> The descriptions are interchanged and corrected. (Line 575 – 576)

Figure 6 caption, lines 572, 576: I would replace “lead-time” with more descriptive “interval used for forecast”

> The description “lead-time” is corrected to “interval used for forecasts” as suggested. (Line 580, 584)

Reviewer #2 (Remarks to the Author):

This is a review of the first revision of an NCOMMS submission that targets predictions of the evolution of tsunamis based on simulated data from a set of virtual observations with a convolutional neural network.

In the revised version, the authors have, most importantly, included an evaluation with real-world observations obtained during the 2011 Tohoku tsunami. This is a meaningful and important addition and helps to make a case for the proposed method. Especially figure 6 is interesting to see. The forecast(naturally) doesn't match the observations exactly, but there's a clear trend and a relatively small offset in the prediction versus the measurement. Additionally, the effect of noise is evaluated with this scenario.

Other improvements include more specifics on error metrics, a discussion of non-earthquake tsunamis, a clarification of the NN architecture and the use of synthetic data (among others).

Overall, I think this version addresses my concerns and I think the manuscript is ready for publication. Especially the addition of the real-world case was good to see and improves the submission substantially.

> We again sincerely appreciate your comments and suggestions which greatly improved the manuscript.

Reviewer #3 (Remarks to the Author):

I feel the authors have improved the ms a lot in this latest revision. Adding the real world example and assessment is very helpful in putting the method and its performance into context.

It seems inevitable that deep learning will become a part of the warning toolkit in natural hazards and tsunami in particular. This study is important as it shows the potential and the difficulty of the approach. It is novel in its particular application.

My initial comments in the first review have been addressed to my satisfaction and the paper reads much better now and is better to understand.

I only have a few suggestions for minor revision. Please consider the following suggested changes:

> Thank you very much again for your careful reading of the manuscript, comments, and suggestions. We have revised the manuscript according to your comments and suggestions as shown below.

L 74: '... predict the tsunami inundation time series at a single location. ...'

> We have corrected the text as suggested. (Line 74)

L 118-120: Here it is not quite clear what 'length' refers to. I am guessing it is the number of samples that are fed into the CNN. The author should say 'number of samples' instead of 'length' (a length is a measure of distance and would require a unit. Here the authors only give a 'number': 3600.

> We agree with your comment. For clarity, we have replaced “length” with “number of samples” and “a data size of 3600.” (Line 120 – 121)

L 266: '... 20 min or less observation time when ...

> We have corrected the text as suggested. (Line 268)

L 313: ... to prepare as many [remove: as possible] tsunami scenarios as possible for Training ...

> We have modified the text as suggested. (Line 318)

L 324: insert the statement in brackets '(Gaussian distribution ...)' [HERE] '... fluctuation [HERE] rather than ...' (L:322)

> We have modified the text as suggested. (Line 327 – 328)